# Linking Soil Fungal Generality to Tree Richness in Young Subtropical Chinese Forests

**DOI:** 10.3390/microorganisms7110547

**Published:** 2019-11-10

**Authors:** Christina Weißbecker, Anna Heintz-Buschart, Helge Bruelheide, François Buscot, Tesfaye Wubet

**Affiliations:** 1Helmholtz-Centre for Environmental Research GmbH - UFZ, Theodor-Lieser-Straße 4, 06120 Halle, Germany; anna.heintz-buschart@ufz.de (A.H.-B.); francois.buscot@ufz.de (F.B.); tesfaye.wubet@ufz.de (T.W.); 2German Centre for Integrative Biodiversity Research (iDiv) Jena-Halle-Leipzig, Deutscher Platz 5e, 04103 Leipzig, Germany; helge.bruelheide@botanik.uni-halle.de; 3Institute of Biology/Geobotany and Botanical Garden, Martin Luther University Halle-Wittenberg, Am Kirchtor 1, 06108 Halle, Germany

**Keywords:** bipartite network, diversity, fungal community assembly, soil, specialization, subtropics

## Abstract

Soil fungi are a highly diverse group of microorganisms that provide many ecosystem services. The mechanisms of soil fungal community assembly must therefore be understood to reliably predict how global changes such as climate warming and biodiversity loss will affect ecosystem functioning. To this end, we assessed fungal communities in experimental subtropical forests by pyrosequencing of the internal transcribed spacer 2 (ITS2) region, and constructed tree-fungal bipartite networks based on the co-occurrence of fungal operational taxonomic units (OTUs) and tree species. The characteristics of the networks and the observed degree of fungal specialization were then analyzed in relation to the level of tree species diversity. Unexpectedly, plots containing two tree species had higher *network connectance* and *fungal generality* values than those with higher tree diversity. Most of the frequent fungal OTUs were saprotrophs. The degree of fungal specialization was highest in tree monocultures. Ectomycorrhizal fungi had higher specialization coefficients than saprotrophic, arbuscular mycorrhizal, and plant pathogenic fungi. High tree species diversity plots with 4 to 16 different tree species sustained the greatest number of fungal species, which is assumed to be beneficial for ecosystem services because it leads to more effective resource exploitation and greater resilience due to functional redundancy.

## 1. Introduction

Soil fungi are a highly diverse group of microorganisms [1,2] that are crucial for soil health [3] and provide many ecosystem services including decomposition, element cycling, plant nutrition, and plant protection [4]. The mechanisms of soil fungal community assembly must therefore be understood to reliably predict how global changes such as climate warming and biodiversity loss will affect ecosystem functionality. Fungal community assembly is influenced by abiotic, biotic factors as well as stochastic and deterministic processes. Key drivers of fungal community composition and richness include soil moisture [5], soil nutrient content [6,7], precipitation [8], and vegetation [9,10]. Tree species loss is a likely consequence of global change, so it is important for silvicultural management to determine how such losses could affect soil fungal communities. We have previously characterized the effects of tree diversity on specific functional groups of soil fungi in subtropical young forests [11]. Here, we extend this analysis by investigating the effects of tree diversity on fungal specialization and tree-fungal network patterns.

Tree species are known to strongly affect ecosystem conditions including soil properties [12,13,14] and microclimate [15,16]. Therefore, regions of high tree diversity have less homogeneous soil and environmental conditions than those with tree monocultures. Additionally, the local conditions in regions of high tree diversity depend strongly on the tree species that are present. The neighborhood conditions of a tree can also result in niche shifts. For example, niche differentiation based on crown height was observed in experimental tree communities with high diversity [17,18]. Similarly, fine root niche complementarity [19] was shown to increase resource capture in mixed stands [20,21], and tree species richness was found to correlate positively with the filling of the soil volume by fine roots [19].

The performance of species under different environmental conditions can differ strongly [22]. Some species can cope with a broad range of environmental conditions and thus occur frequently in many different habitats. Other species, known as specialists, only perform well in a narrow range of environmental conditions. Therefore, it is assumed that well-adapted specialists will outperform generalists in homogenous environments, while the reverse will be true in heterogeneous environments. In molecular soil fungal ecology, the abundance of a fungal taxon can be regarded as a proxy for its performance because it is assumed that well-performing species will be more able to proliferate and will thus have a greater chance of being detected.

Network analysis is a technique that originated in the social sciences but has been widely used in community macroecology, for instance to characterize pollinator-plant or predator-prey interaction networks [23,24]. The advent of molecular high throughput sequencing technologies has enabled this technique to also be used in microbiology to clarify the mechanisms that structure fungal communities in a way that complements descriptive investigations based on alpha and beta diversity relationships [25,26,27]. Network analysis can be used to assess the ecological interactions between functionally different partners and to deduce their ecosystem-level consequences in a more integrated manner than is possible by intraspecific investigation. Network analyses inherently account for the fact that all components of an ecosystem are interconnected [28]. Consequently, ecological network analyses are increasingly being used to evaluate the effects of environmental change on ecosystems [29,30]. For example, Tylianakis*,* et al. [31] found that anthropogenic habitat modification did not affect species richness but significantly influenced the network structure of bees, wasps, and their parasitoids, affecting parasitism rates and thus pollination. Plant-fungal networks have been analyzed to support or better understand disease management [32], ecosystem development [27], succession and seasonality [33], latitudinal gradients [34], and host preferences [35,36,37,38,39]. However, to our knowledge, this work is the first to examine the effects of tree species diversity on tree-soil fungal network structure and soil fungal specialization, and the likely consequences of global tree species loss. The data analyzed here were derived from the Biodiversity and Ecosystem Functioning experiment China (BEF China) [40,41], which features plots having one, two, four, eight, and 16 different tree species.

We performed a tree-fungal bipartite network analysis using a subsampling approach and evaluated the network metrics specified below in relation to three tree diversity levels. Additionally, we analyzed the specialization of fungal OTUs based on the phi coefficient [42] and assessed differences in the specialization of specific fungal functional groups. Our analysis is based on several network structure metrics, including the main metrics of *nestedness*, *modularity*, and *connectance*, as well as *generality*—a measure of network asymmetry (Appendix A). The latter metric was included because studies on consumer-prey networks have shown that environmental change can affect consumer-prey asymmetries without strongly affecting other network metrics [29]. The fungal *C score*, as measure of disaggregation, was also computed to deduce possible mechanisms of fungal community assembly [43,44].

*Nestedness* measures the extent to which specialist species of higher trophic levels (e.g., pollinators) interact with generalist species of lower trophic levels (e.g., plants). Each generalist species typically interacts with many higher trophic level species [30]. Highly nested communities are assumed to be stable because most of the interactions involve generalist species, so the overall network structure will not be greatly affected if a disturbance removes a specialist species. Non-nested patterns may be either modular or checkerboard (anti-nested). In modular patterns, there are sets of species that interact more strongly with one-another than with species outside the set. Such patterns may result from evolutionary processes that favor the emergence of highly co-adapted species (for example, species that form symbiotic interactions [45]) or modules that independently perform specialized functions [46]. To assess general network structure in terms of nestedness, computed nestedness values were compared to those for a randomized dataset (a null model). *Connectance* is the ratio of the number of interactions in the network to the total number of possible interactions, while fungal *generality* is defined as the mean number of tree species per fungal species.

We hypothesized that increasing tree diversity would increase connectance and fungal generality while reducing modularity, fungal C score, and fungal specialization as measured by the phi coefficient.

## 2. Materials and Methods

We used previously published amplicon sequencing data to construct interaction networks [11]. For details of the soil sampling, soil sample preparation, nucleic acid extraction, 454 pyrosequencing, and bioinformatics analysis procedures, please see the work of Weißbecker, Wubet, Lentendu, Kühn, Scholten, Bruelheide and Buscot [11]. Here we briefly outline the experimental design and major sample processing procedures, and describe in detail the data processing steps involved in the network and statistical analyses.

### 2.1. Sampling Site

Our study was conducted in the frame of the Biodiversity and Ecosystem Functioning experiment China (BEF China [40]). In 2009, experimental forest plots were established on a hillside in Jiangxi Province in Southeastern China (29 °C07′26.0″ N 117 °C54′29.0″ E). The site’s climate is subtropical, with warm wet summers and cold dry winters. A broken-stick design was used to determine the experimental planting schemes of the 31 forest plots investigated here: a set of 16 native subtropical tree species was repeatedly sub-divided into subsets of eight, four, two and one species to establish communities with lower tree diversity levels (Appendix A). The total species pool had equal numbers of arbuscular mycorrhizal- (AM) and ectomycorrhizal- (EcM) forming tree species. Each forest plot covered 25.8 m × 25.8 m. In each plot, 400 trees were planted with a spacing of 1.29 m. In October 2011, the mean total tree height ranged from 52 to 301 cm depending on tree species [47].

### 2.2. Soil Sampling

In October 2011, we randomly selected five tree individuals per tree species in each plot (where possible) for root zone sampling, which was performed by using an augur to remove four soil cores (6 cm in diameter and 10 cm deep) at points 20–30 cm from the tree trunk in each of the cardinal compass directions. The four soil cores were then mixed, sieved (2 mm mesh size), and homogenized to form a composite soil sample. The experimental plots were planted according to a broken-stick design (Appendix A), and the number of experimental plots chosen for sampling decreased with increasing tree diversity while the number of samples collected per plot increased (Appendix A). Two 15 g subsamples from each pooled sample were immediately flash-frozen in liquid nitrogen. One subsample was then freeze-dried [48] and transported by airplane within 4 days to the processing lab in Germany, where it was immediately stored at −80 °C until needed for molecular analysis.

### 2.3. Nucleic Acid Extraction and Multiplexed Amplicon Pyrosequencing

Microbial DNA was extracted with a PowerSoil^®^ htp 96 Well Soil DNA Isolation Kit or a PowerSoil^®^ Total RNA Isolation Kit (MO BIO Laboratories Inc., Carlsbad, CA, USA) in combination with a PowerSoil^®^ DNA Elution Accessory Kit. Fungal ITS rDNA amplicon libraries were generated using the fungal-specific ITS1f primer [49] containing Roche 454 pyrosequencing adaptor B, the universal ITS4 [50], Roche 454 pyrosequencing adaptor A, and a sample-specific multiplex identifier sequence (MID). All samples were subjected to three replicate PCR reactions. PCR products were cleaned, quantified, and processed using the GS FLX+ sequencing kit (Roche, Mannheim, Germany). The amplicons were sequenced by unidirectional pyrosequencing from the ITS4 ends using a Roche GS FLX+ 454 pyrosequencer at the Department of Soil Ecology, Helmholtz Centre of Environmental Research (UFZ, Halle, Germany).

### 2.4. Bioinformatic Analysis

Multiple levels of sequence processing and quality filtering were applied using an in-house metabarcode analysis pipeline for grid engines based mainly on the MOTHUR [51] and OBITools [52] software suites. Sequences with ambiguous bases, barcode mismatches, or homopolymers exceeding eight nucleotides were discarded. FlowClus [53] was used to denoise flows and trim reads into uniform 360 bp long read fragments spanning the ITS2 region and the 5.8S rRNA gene. Chimeric reads were removed using UCHIME [54] and quality filtered sequences were clustered into operational taxonomic units (OTUs) using vsearch [55] with a sequence similarity threshold of 97%. OTUs were taxonomically assigned using the UNITE database version v7_2 [56]. Putative functions were annotated using the FUNGuild fungal database [57].

### 2.5. Data Processing

Data processing and statistical analyses were performed using R (version 3.5.2, [58]). The phyloseq package [59] was used to combine and process OTU count and environmental data. Rare fungal OTUs comprising only singleton, doubleton, and tripleton sequences were discarded [60]. Sequences were rarefied to 700 sequences per sample. All remaining OTUs with at least 10 sequences in the total rarefied dataset were considered in subsequent analyses [32]. The abundance data were transformed into incidence data. Other R packages used for data management and visualization included BiocManager [61], biomformat [62], dplyr [63], data.table [64], extrafont [65], gdata [66], ggplot2 [67], plyr [68], prodlim [69] and vegan [70].

### 2.6. Tree-Fungal Bipartite Analysis in a Subsampling Approach

We performed a fungal-tree bipartite network analysis based on observations of fungal-tree co-occurrence using the bipartite package [71]. In accordance with our sampling design, we sampled each of the 31 forest plots of the broken-stick design.

No replicates of tree species mixtures were sampled. For each tree species, we collected five samples at each diversity level. The number of collected samples per plot thus increased with the diversity level: five samples were collected from each monoculture plot, whereas 80 (16 × 5) samples were collected from the 16 tree species mixture plot. Because the number of forest plots decreased as the tree diversity level increased, we aggregated the data for the four, eight and 16 tree species mixture plots into a single “high tree diversity” dataset (Figure 1a). Thus, the “high tree diversity dataset” represented seven independent forest plots compared to eight two-tree species mixtures and 16 tree monoculture plots. We therefore constructed our bipartite networks (see Figure 1c) using a subsampling approach in which each subsample was based on seven plots per tree species diversity level and seven tree species. This ensured that all networks were based on the same number of individual plots and the same number of samples within a plot.

Within a given subsampling combination, the same seven tree species were investigated at all three diversity levels and only one tree species was sampled per plot (Figure 1b). For the two tree species mixtures, there were 1024 (8 × 2^7^) valid subsamples based on seven independent plots with one tree species per plot. However, the tree species *Castanopsis eyrei* suffered severe mortality and comparatively few individuals of this species were planted initially. Therefore, at the time of sampling, only a few individuals of this species remained in the experiment, so it was excluded from our analysis. Consequently, there were 576 independent combinations of seven tree species and seven two-species plots that could be used to generate bipartite networks. Bipartite networks were generated based on tree-fungal co-occurrence (Figure 1c) for each of the possible co-occurrence thresholds. That is to say, networks were generated based on the observation of tree-fungus co-occurrence in one, two, three, four or five of the five soil samples collected for each tree species at each diversity level. We only considered presence-absence data. Network topological characteristics (Figure 1d) were calculated at the network and fungal OTU levels using the networklevel and grouplevel functions of the bipartite package, respectively. For each tree diversity level, we calculated the fungal richness, Shannon diversity, and the following network characteristics: number of fungal OTUs, nestedness (NODF), network connectance, fungal generality, mean number of shared fungal partners, and fungal C score. The Kruskal-Wallis test was used to assess the statistical significance of differences in network characteristics between tree diversity levels based on the 576 data points generated by the subsampling approach. The Kruskal-Wallis test for multiple comparisons (as implemented in the pigrmess package [72]) was used as a post hoc test to perform pairwise group comparisons between the three tree diversity levels.

According to Almeida-Neto*,* et al. [73] the nestedness metric Nestedness metric based on Overlap and Decreasing Fill (NODF) is more robust than the nested temperature metric; higher NODF values indicate greater nestedness. NODF values of our data were statistically compared them to NODF values generated using a simulated null model. The null model was created by shuffling the OTU abundance data before it was divided into subsets corresponding to different tree diversity levels. The column and row sums of the data were kept constant during shuffling. We then used the nullmodel function of the vegan package with the “r2dtable” method to create the null models.

### 2.7. Specialization Analysis

To complement the bipartite network analysis, we assessed the degree of fungal specialization across the tree diversity levels and among the fungal functional groups. The specialization of each fungal OTU for each tree species was assessed by computing the φ (phi) specialization coefficient based on presence/absence data using Equation (1) [42]:φ = ± √(X^2^/N) = (a × d−b × c)/√((a + b) × (c + d) × (a + c) × (b + d))(1)
where X^2^ is the chi-square statistic for a 2 × 2 contingency table with N being the total number of observations, a the number of occurrences of a fungal OTU in a plot containing a particular tree species, b the number of occurrences in plots without that species, c the number of times the fungal OTU is absent in plots containing that species, and d the number of times the fungal OTU is absent in all other plots. The phi coefficient ranges from −1 to 1; the extrema of this range indicate a fungal OTU that always avoids the tree species in question and one that is only found in association with that tree species, respectively.

We determined the median phi coefficient for each of the 576 subsampling combinations (see Section 2.7), generating seven plots for each of the three tree diversity levels. The median value of the tree-specific positive phi coefficients of the present OTUs was then calculated for each subsampling combination. Boxplots were used to visualize the median phi coefficients of the subsampling combinations for each tree diversity level. We also determined whether the phi coefficient differed between fungal functional groups and analyzed the differences in the calculated positive phi values. The Kruskal-Wallis test and the Kruskal-Wallis test for multiple comparisons with the Bonferroni correction were used to assess the statistical significance of observed differences, as the subsampling approach did not commonly yield normally distributed results. For each fungal OTU, we calculated the maximum phi coefficient across all tree species and identified the 200 fungal OTUs with the highest maximum phi coefficients. The phi coefficients of these fungal OTUs were visualized in a heatmap and clustered using Euclidean distance-based hierarchical clustering dendrograms. The R packages used for this purpose were gplots [66], colorspace [74], and dendextend [75]. We also determined the taxonomic identities of the 20 fungal OTUs with the highest positive phi coefficients. To complement the specialization pattern analysis, we also assessed the taxonomic identity of the most frequent fungal species in all the subsampling combinations. We defined a fungal OTU as being frequent if it occurred in all seven plots of at least one subsampling combination. All fungal OTUs showing this high occurrence pattern at all three diversity levels in at least one subsampling combination were identified taxonomically. In addition, we identified all of the fungal OTUs that were only frequent at one diversity level (which we termed “unique frequent fungal OTUs”) and investigated their occurrence patterns at the diversity levels in which they were not frequent.

## 3. Results

Taxonomic assignments of fungal OTUs, the assignments of OTUs to functional groups, and the effects of environmental, spatial, and biotic factors on fungal community composition and diversity were reported by Weißbecker, Wubet, Lentendu, Kühn, Scholten, Bruelheide and Buscot [11]. Briefly, pyrosequencing generated 1,155,299 raw sequences (737,907 sequences after quality filtering and removal of OTUs with less than four reads) from the 394 collected soil samples. Among the major fungal functional groups, saprotrophic fungi dominated, accounting for 31% of the detected OTUs. Less common functional groups were EcM fungi (7% of all OTUs), AM fungi (5%), and plant pathogens (5%); 46% of the fungal OTUs could not be assigned to a functional group. The final dataset for the following analyses (rarefied to a uniform number of 700 sequences per sample and pruned to exclude OTUs not containing at least 10 sequence reads) comprised 248,026 sequences that were clustered into 1926 fungal OTUs. The analysis was based on three data subsets representing: (i) tree monoculture plots (ii) two tree species mixture plots, and (iii) high tree diversity plots (i.e., plots with 4, 8, or 16 tree species). Rarefaction curves for these data subsets are shown in Appendix A.

### 3.1. Tree-Fungal Bipartite Network Analysis with a Subsampling Approach

The network analysis was based on a subsampling approach (see Method Section 2.7 and Figure 1), which was used to generate all the results presented below. The topological characteristics of the tree-fungal bipartite network were calculated at the network and group levels for all possible tree-fungal co-occurrence link thresholds (Appendix A). Although the specific values of the network parameters depended on the choice of link threshold, the general trends between tree diversity levels were robust (Figure 2). Increasing the link threshold generally reduced the number of fungal OTUs retained in the bipartite networks (Figure 2a) from about 1000 fungal OTUs for a threshold of 1/5 to about 50 OTUs for a threshold of 5/5. Table 1 presents the full set of results obtained using a link threshold of 3/5 (meaning that the bipartite network only included a link between an OTU and a tree species if at least three of the five samples collected for that tree species showed the presence of that fungal OTU). The fungal richness, fungal Shannon diversity, and fungal C score for the monocultures and the two tree species mixtures did not differ significantly but were significantly lower than those for the high tree diversity mixtures (Table 1, Appendix A). At the network level, we analyzed nestedness, network modularity, and network connectance. Tree-fungal networks were less nested (i.e., had lower NODF values) than the null model (Table 2). The two tree species diversity level had the lowest network modularity value and the highest network connectance and fungal generality. All calculated networks consisted of a single module.

### 3.2. Fungal Specialization Patterns as Evaluated using the Phi Coefficient

The specialization of the fungal community at the three tree diversity levels was assessed by computing the median phi coefficients for the 576 subsampling combinations. Specialization was lowest in the two tree species mixtures plots and highest in the tree monocultures (Figure 3). The EcM fungi exhibited a greater degree of specialization than the other fungal functional groups (Figure 4; a table showing the phi coefficients of all the fungal OTUs is available at the zenodo archive); the degrees of specialization of the other groups (saprotrophs, plant pathogens, and AM fungi) did not differ significantly. Additionally, the degree of specialization of saprotrophic fungi in plots with AM tree species was significantly higher than in those with EcM tree species (data not shown). We visualized the distributions of the 200 most specialized fungal OTUs in a heatmap covering all the studied tree species (Figure 5), indicating different numbers of specialized OTUs per tree species. Taxonomic identifications of the 20 most highly specialized fungal OTUs are presented in Appendix A; eight of these OTUs were EcM fungi, four were saprotrophs, one was an orchid mycorrhizal OTU, and seven belonged to unknown fungal functional groups. Fifteen fungal OTUs were identified as frequent fungal species at all three tree diversity levels (Appendix A). Most (nine) of these frequent fungal OTUs were saprotrophs (nine OTUs), but two were plant pathogens and one arbuscular mycorrhizal fungal OTU was also identified.

All fungal taxa that were frequent at only one tree diversity level also occurred at the other tree diversity levels at lower frequencies (see histograms indicating frequency of observation in 1–7 trees in the subsampling approach in Appendix A).

## 4. Discussion

In this study, we analyzed the relationship between tree diversity, tree-fungal bipartite network structure, and fungal specialization in young subtropical forest plantations. Weißbecker et al. [11] previously found that local tree species richness had no effect on soil fungal OTU richness. Here, using a network analysis approach that combines tree diversity levels, we found that plots with high tree species diversity (i.e., plots containing four to 16 different tree species) exhibited increased fungal diversity. We also observed differences in the network structure of fungal-tree bipartite networks and differences in the degree of fungal specialization between tree diversity levels.

### 4.1. Increased Fungal Alpha Diversity in Plots with High Tree Species Diversity

The fungal alpha diversity (richness and Shannon diversity) was significantly greater for the high diversity tree species mixtures than for tree monocultures and two tree species mixtures. Tree species richness enhances forest productivity [76,77,78] and can thus yield higher productivity compared to monocultures (overyielding). Therefore, in addition to providing a greater variety of distinct niches, increasing tree species diversity could increase the quantity of resources (e.g., rhizodeposits, litter input, and fine root turnover) available to fungi, thereby increasing the fungal diversity that can be sustained. A more diverse fungal community might also enhance tree productivity; the two effects could thus be complementary. Accordingly, in a separate study conducted at the site considered here, Fichtner*,* et al. [79] found that local neighborhood tree species richness increased tree community productivity due to facilitation and competitive reduction. Our previous study [11] revealed no comparable positive effects of tree species diversity on fungal richness. However, that study was conducted at the local neighborhood scale, with tree species diversity values ranging from one to eight because only one focal sampling tree and its eight nearest neighbors were considered. In this work, we instead focused on abundant fungal OTUs (i.e., those represented by at least 10 sequence reads) and binned data representing five diversity levels into three wider diversity categories, increasing the statistical power of our analysis. This resulted in the detection of a positive effect of tree species richness on the abundance of fungal taxa.

### 4.2. The Connectance and Fungal Generality of Tree-Fungal Bipartite Networks are Highest at the Two Tree Species Diversity Level

Next to the relationship between fungal and tree species diversity, we investigated tree species –fungal OTU co-occurrence patterns. The computed network characteristics revealed significant differences between the low tree diversity plots (monocultures and two tree species mixtures) and those with high tree diversity (four, eight or 16 tree species, see Table 1). Independently of the chosen link threshold, none of the network characteristics of the tree-fungal bipartite networks differed significantly between the monocultures and the two tree species mixtures. However, contradicting our hypotheses that network connectance and generality would increase and modularity and C scores would decrease with tree diversity, we found that the high tree diversity plots had: (i) lower median connectance and fungal generality values than the monoculture and two tree species mixture plots as well as (ii) higher modularity values and fungal C scores.

Our hypothesis that fungal specialization would decrease with tree diversity was supported by the finding that monoculture plots had the highest degree of fungal specialization, which suggests that fungal specialists outcompete generalists in the relatively homogeneous environments created when only one tree species is present. The two tree species networks had higher network connectance and a lower degree of fungal specialization than those for monocultures. These observations also support our hypothesis that generalist fungi can cope adequately with the more heterogeneous environments created by the presence of two tree species, and outcompete specialist fungi that only perform well in one of the two niches created by the two tree species. However, our initial hypothesis was not supported by the finding that high tree diversity mixtures had a greater degree of specialization and lower network connectance than the two tree species mixtures. Planting several tree species together presumably creates more environmental niches than are present in monocultures due to both species diversity and interaction effects/processes. Additionally, highly diverse tree species mixtures may offer habitats suitable for fungi specialized in connecting different tree species, i.e., those fungi that need resources from different trees to which they are connected.

Frequent species are believed to provide crucial network structure support and resilience [80] because they are not limited by resource or partner availability [81]. Therefore, a high number of frequent species is sometimes taken as an indicator of ecosystem stability. On the other hand, specialist species broaden functionality and resource use. Since silvicultural practice affects tree diversity [82] and the associate fungal communities in terms of diversity and homogenecity [83], of the three tree species diversity levels considered in this work, the high diversity level may be ecologically preferable in terms of fungal richness and the number of specialist and frequent taxa for three reasons. First, it has the highest number of fungal species. Second, these fungal species include more specialized fungal taxa than are present in plots with less diverse tree species mixtures, meaning that the fungal community’s resource usage is broader. Third, the frequent species found at the lower tree diversity levels are also present at the high diversity level, albeit at reduced frequencies. In the event of tree diversity loss from the high tree diversity plots (which could cause the loss of some specialized fungi), these fungal species may increase in abundance and become frequent, taking over ecological processes such as decomposition, tree protection, and tree nutrition.

EcM fungal OTUs exhibited a significantly higher degree of specialization than the other fungal functional groups (saprotrophic fungi, AM fungi, and plant pathogens). Moreover, unlike other fungal functional groups, EcM fungal communities reportedly exhibit significant host effects [11]. In general, evolutionary history suggests that EcM fungi are more highly specialized than AM fungi [84]. The number of plant host species and EcM fungal species is similar (about 6000 species) [1,85,86,87], while AM plants comprise around 80% of all plant species [1,88] but only around 300 AM fungal morphospecies have been described [89]. Nevertheless, some degree of host preference has been reported for AM fungi [90,91,92]. Bennett*,* et al. [33] found a higher degree of specialization in tree-AM fungal networks of old forests (>130 years) than in young forests (25 years), and proposed that specialization in AM fungi will become more pronounced as a forest develops after clear-cutting. These authors also suggested that post-disturbance (clear-cutting) associations might reflect the local availability of fungal taxa rather than the intrinsic host preferences of AM fungi.

### 4.3. Comparison with Other Bipartite Network Studies

In general, our tree-fungal bipartite networks exhibited low to moderate network connectance (0.20–0.27), high modularity (0.41–0.58), and non-nested structures. For comparative purposes, the network characteristics determined in other plant-soil fungal studies are presented in Table 3. Network metrics depend strongly on the number of nodes included, and care must be taken when comparing network metrics from different studies. For example Fodor [44] found a high network connectance in mature forests (55–100 years old) and concluded that mycorrhizal fungi (which were predominantly generalists) acted as connector organisms linking the tree species. This pattern did not exist in our young forests (which were sampled in the third growing season after planting) even though many EcM tree species and tree individuals were present. The fungal communities at our sampling site were characterized by limited dispersal and a high beta-diversity across and within plots [11]. This indicates that mycorrhizal networks had not yet been established at the plot scale; the fungal communities (especially those of EcM fungi) differed strongly between samples within the same plot [11]. Whereas the tree-EcM networks analyzed by Fodor [44] showed a nested pattern, Bahram*,* et al. [93] found tree-EcM networks to be non-nested. While some studies suggest that mutualistic networks have inherently nested structures [94], nestedness patterns in soil fungal communities span the full spectrum of possible structures, ranging from nested [25,44,94] to non-nested [95] and even anti-nested [34] (Table 2). Nestedness is a core network metric because it has been suggested to be related to network persistence [96].

In contrast to previous bipartite network studies, all our networks consisted of a single module with high modularity, “indicating the possible presence of community structure” [97]. A modular structure indicates that groups of nodes perform different functions with some independency from one-another [46]. For example, Toju et al. [95] found eight interconnected modules with differing fungal functional group compositions. The high modularity [98] of our networks (>0.4) may indicate the presence of different fungal functional groups that assemble in different ways relative to the tree community [11]. The computational method used in this work only divides networks into multiple modules if the number of edges between communities is lower than expected [97], which was not the case for our networks. The low number of modules per network (one) may be another indicator of the early developmental state of our networks, indicating that they have yet to form densely interconnected modules.

## 5. Conclusions

In accordance with our hypothesis, tree monocultures had the highest frequencies of specialist fungi. However, the degree of fungal specialization and network segregation were higher in plots with high tree diversity than in those with only two tree species. There is ongoing and global interest in clarifying the impact of tree diversity on sustainable forest plantations [99,100]. Plots with high tree diversity (i.e., those with four to 16 different tree species) supported the greatest number of fungal species, which is assumed to be beneficial for ecosystem service provision because greater fungal diversity enables more effective resource exploitation and confers greater resilience due to functional redundancy.

## Figures and Tables

**Figure 1 microorganisms-07-00547-f001:**
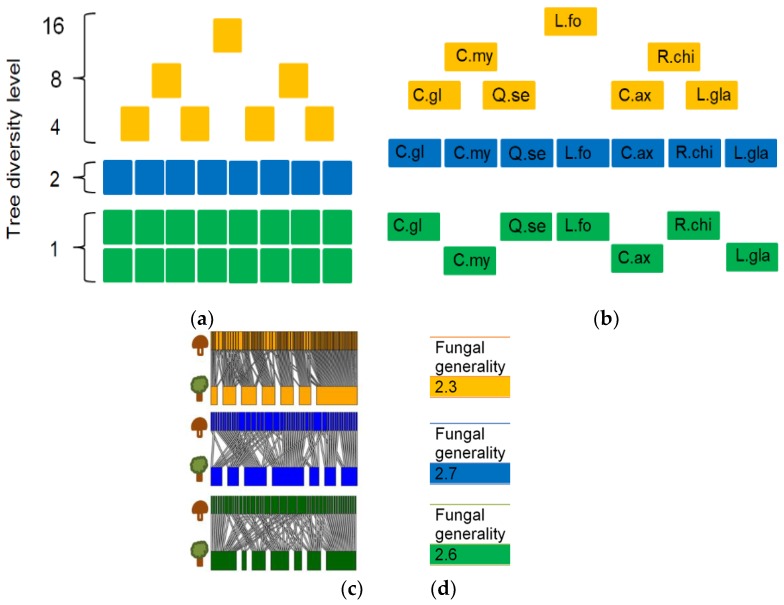
The bipartite network analysis procedure. Data were pooled into three tree diversity levels (**a**). An illustrative subsampling set (**b**). For each subsampling combination, a bipartite network was generated (**c**) and network characteristics such as fungal generality were computed (**d**). Statistical differences between the tree diversity levels could be analyzed by considering the combined network characteristics of 576 possible subsampling combinations at each tree species diversity level.

**Figure 2 microorganisms-07-00547-f002:**
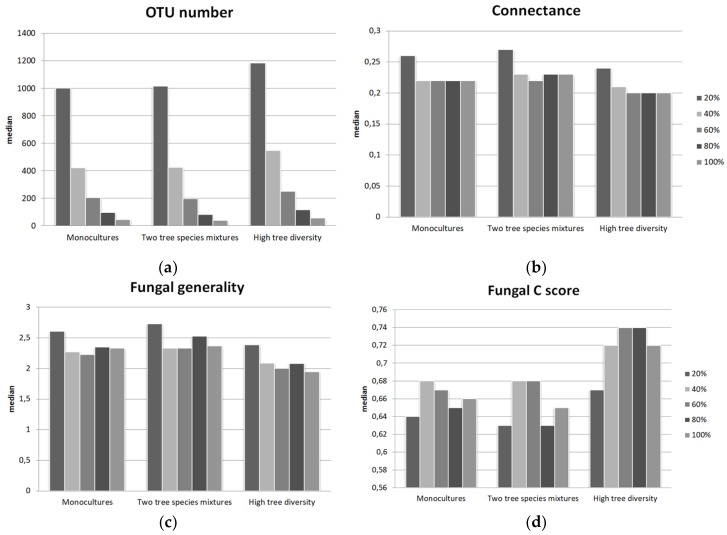
Dependence of the calculated network characteristics on the link threshold for tree species –fungal OTU co-occurrence in the bipartite network analysis. The charts show the median values (based on 576 subsamples) of four key network characteristics: fungal OTU number (**a**), network connectance (**b**), fungal generality (**c**) and fungal C score (**d**). A table showing all of the computed network characteristics is available in the Appendix A.

**Figure 3 microorganisms-07-00547-f003:**
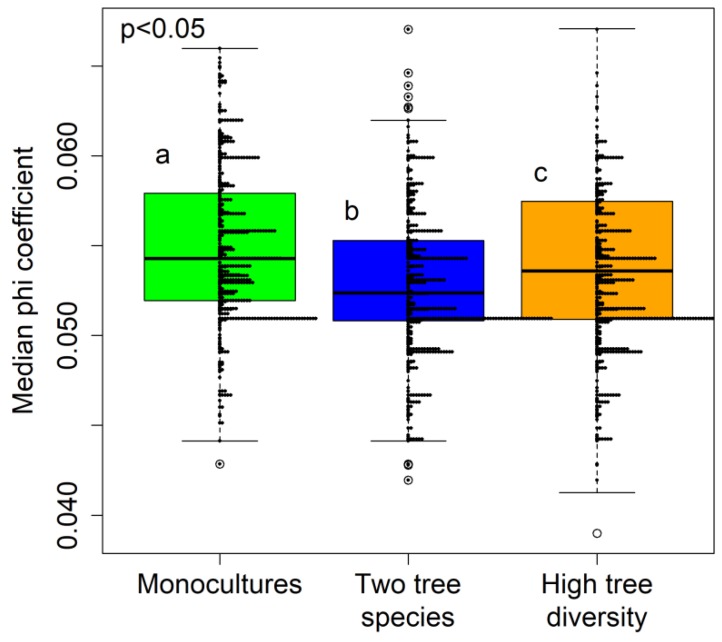
Boxplots of phi coefficients for the three tree diversity levels, with histogram of the results of all subsampling combinations indicated as horizontal arrays. The Kruskal-Wallis rank sum test and Kruskal-Wallis test for multiple comparisons were used to evaluate the significance of group differences; a, b, and c indicate significantly different groups.

**Figure 4 microorganisms-07-00547-f004:**
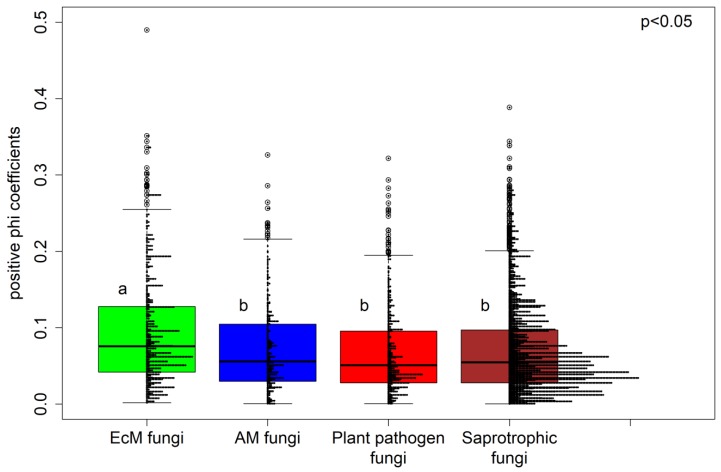
Boxplot showing phi specialization coefficients for the main fungal functional groups, with histogram of the results of all subsampling combinations indicated as horizontal arrays; a and b indicate significantly different groups.

**Figure 5 microorganisms-07-00547-f005:**
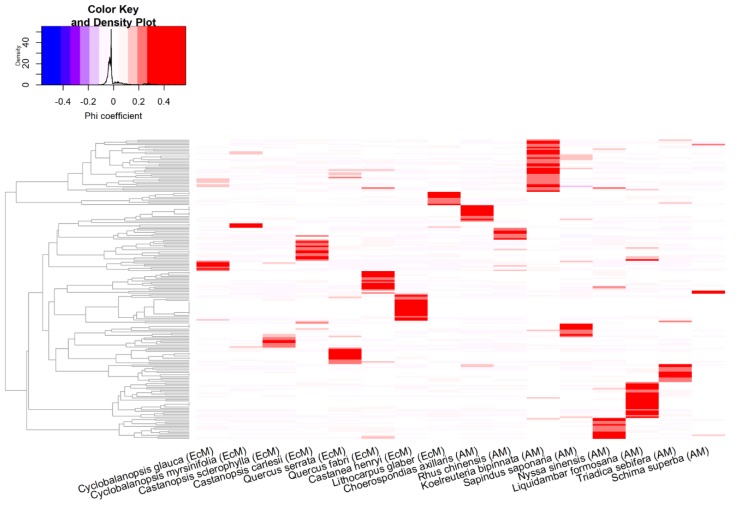
Heat map showing the distribution patterns of the 200 fungal OTUs with the highest phi specialization coefficients, ordered according to a hierarchical clustering of the phi coefficients, among the 16 tree species.

**Table 1 microorganisms-07-00547-t001:** Calculated network metrics for different levels of tree species diversity based on 576 tree-fungal bipartite subsamples and three tree diversity levels: monocultures (“Mono”), two-tree species mixtures (“Two mix.”) and high tree diversity mixtures (“High”). The Kruskal-Wallis test was used to identify significant differences in network values across the tree diversity levels. The median values of the network characteristics are reported for each tree diversity level. The Kruskal-Wallis test for multiple comparisons with the Bonferroni-Holmes correction was used to assess the significance of pairwise differences in network characteristics across tree diversity levels (n.s.: no significant difference detected). Numbers indicate the tree diversity levels: 1-monocultures, 2-two tree species mixtures, 3-high diversity tree species mixtures. Results are shown for networks generated using a tree species-fungal OTU co-occurrence threshold of 3/5.

	Number of OTUs in Network	Modularity	Connectance ^1^	Fungal Generality ^2^	Fungal C Score ^3^	Mean Number of Shared Fungal Partners ^4^	Fungal OTU Richness	Fungal Shannon Diversity
Kruskal p	<0.001	<0.001	<0.001	<0.001	<0.001	<0.001	<0.001	<0.001
Median								
Mono	206	0.52	0.22	2.26	0.67	9.43	1004	4.99
Two mix.	198	0.51	0.22	2.33	0.68	9.57	1017	5.06
High	251	0.58	0.2	2	0.74	8.48	1187	5.34
Pairwise p								
1-2	<0.001	n.s.	n.s.	<0.001	n.s.	n.s.	n.s.	n.s.
1-3	<0.001	<0.001	<0.001	<0.001	<0.001	<0.001	<0.001	<0.001
2-3	<0.001	<0.001	<0.001	<0.001	<0.001	<0.001	<0.001	<0.001

^1^ Network connectance: Realized proportion of possible links, ^2^ Fungal generality: Mean effective number of tree species per fungal species, ^3^ Fungal C score: Average degree of co-occurrence for all possible pairs of fungal OTUs. Values close to 1 indicate evidence for disaggregation, e.g., through competition. Values close to 0 indicate aggregation of species (i.e., no repelling forces between species), ^4^ Mean number of shared fungal partners: Mean number of fungal species that interact with at least two tree species

**Table 2 microorganisms-07-00547-t002:** Median nestedness (NODF) values for three tree diversity levels (monocultures, two tree species mixtures, and high tree diversity mixtures) based on null models and bipartite networks generated for 576 subsamples. Networks were generated using a tree species-fungal OTU co-occurrence threshold of 3/5.

Networks	NODF Median	Wilcox.p
Tree monocultures	21.51	<0.001
Null model	57.6	
Two tree species mixtures	22.59	<0.001
Null model	57.48	
High tree species mixtures	15.66	<0.001
Null model	57.32	

**Table 3 microorganisms-07-00547-t003:** Network metrics reported in previously published plant-fungal network studies.

	This Study	[94]	[34]	[95]	[44]	[33] *	[26] *	[25]
Study system	16 subtropical tree species in a forest biodiversity experiment	Semi natural grasslands, 33 plant species	cool-temperate, warm-temperate and subtropical forests	Temperate forest with 33 tree species	Temperate forests, mainly *Quercus* and *Carpinus*	33 understory plant species in temperate spruce forest		Xeric shrubland
Country	China	Estonia	Japan	Japan	Romania	Estonia		Mexico
Age	3 years				55–100 years	25 years and 130 years	130 years	
Treatment	Tree species diversity	Host plant functional group	Latitudinal gradient			Succession and seasonality		
Samples	Soil within tree rooting zone	Root samples	Root samples	Root samples	aboveground EcM fructifications	Root samples		Root samples
Study target	Soil fungi	AM fungi	Soil fungi, fungal groups	Soil fungi	EcM fungi	AM fungi		AM fungi
Nestedness	Less nested (15.66–29.42,) than random (53.87–60.04) NODF	More nested than random (27) nestedness temperature)	Anti-nested (−9 to −4) weighted NODF)	Less nested (25–35,) than random (32–40) weighted NODF	More nested (16) than random (38, 31) nestedness temperature)			More nested (14.36–54.83) than random, NODF
Number of modules	1	5		8	4		5-9	
Modularity	0.41–0.58	Higher than random 0.18		Moderate to low modularity (0.35–0.42), higher than random (0.32–0.38)	Low modularity 0.24	0.3–0.44		Modular 0.30–0.57
Connectance	0.20–0.27	Less connected than random 0.52	0.07	0.1-0.55	High connectance 0.42			Low connectance 0.05–0.15
Fungal generality	1.95–2.73					2.25–4.0		
Fungal C score	0.63–0.74				No difference of observed (0.59) and random value (0.58)			

* these studies re-evaluated the data from [91]; empty fields indicate no available information.

## Data Availability

Relevant materials and protocols will be made available upon request. Datasets of the raw sequences generated for this study can be found in the European Nucleotide Archive (https://www.ebi.ac.uk/ena/data/view/PRJEB12020) [101]. The bioinformatically processed sequence dataset and metadata can be found in the Zenodo repository (https://zenodo.org/record/1215505) [102]. The R scripts generated for the statistical analyses and the table with the fungal OTU phi coefficients are available in a Zenodo repository (https://zenodo.org/record/3533732) [103].

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
