# Peer review of "Linking Soil Fungal Generality to Tree Richness in Young Subtropical Chinese Forests"

_microorganisms, 2019, doi:10.3390/microorganisms7110547_

Round 1

Reviewer 1 Report

The paper is valuable contribution to the knowledge on the soil fungal community and fungal specialization. The authors make a very good job and the manuscript is very well worth of publishing. I have only a few comments, listed below.

Line 37: I`d suggest - abiotic, biotic factors as well as deteministic and stochastic processes

Line 44: Please add smth about effects of silvicultural management on the tree diversity (F., Sánchez, M., del Río, M., & Cañellas, I. (2005). Using historic management records to characterize the effects of management on the structural diversity of forests. Forest Ecology and Management, 207(1-2), 279-293.

Line 79-87: I`d move this paragraph (or most part) to MM

Line 245-248 - Could you add number of sequences passing quality filtering, number of excluded singleton abd number of clusters (non-singletons)

Lin4 360-370 - Could you also discuss here possible biodiversity differences in fungal community between managed and unmanaged forests/ disturbed forest? I mean that fungal communities in disturbed forest/managed forest are more homogeneous and diverse than in areas subject to light logging – (Bachelot, B., Uriarte, M., Zimmerman, J. K., Thompson, J., Leff, J. W., Asiaii, A., ... & McGuire, K. (2016). Long‐lasting effects of land use history on soil fungal communities in second‐growth tropical rain forests. Ecological applications, 26(6), 1881-1895).

Author Response

Note: Our replies to the reviewer comments are set in italic characters.

Reviewer 1

The paper is valuable contribution to the knowledge on the soil fungal community and fungal specialization. The authors make a very good job and the manuscript is very well worth of publishing. I have only a few comments, listed below.

We would like to thank you for the appreciation of our manuscript and your constructive suggestions. We have addressed all your concerns. Please find the responses point-by-point below.

Line 37: I`d suggest - abiotic, biotic factors as well as deteministic and stochastic processes.

Thank you for the suggestion. We have edited the sentence accordingly.

Line 44: Please add smth about effects of silvicultural management on the tree diversity (F., Sánchez, M., del Río, M., & Cañellas, I. (2005). Using historic management records to characterize the effects of management on the structural diversity of forests. Forest Ecology and Management, 207(1-2), 279-293.

Thank you for this comment. We have included the reference to silviculture here and discuss the mentioned papers together with your other suggestion in the discussion (l. 515 in the track-changes version).

Line 79-87: I`d move this paragraph (or most part) to MM

We have kept this paragraph in the introduction in response to the other reviewer who has requested to make this part of the introduction more detailed.  

Line 245-248 - Could you add number of sequences passing quality filtering, number of excluded singleton abd number of clusters (non-singletons)

We have included the number of finally accepted reads, as suggested (l. 346 in the track-changes version).

*Lin4 360-370 - Could you also discuss here possible biodiversity differences in fungal community between managed and unmanaged forests/ disturbed forest? I mean that fungal communities in disturbed forest/managed forest are more homogeneous and diverse than in areas subject to light logging – (Bachelot, B., Uriarte, M., Zimmerman, J. K., Thompson, J., Leff, J. W., Asiaii, A., ... & McGuire, K. (2016). Long‐lasting effects of land use history on soil fungal communities in second‐growth tropical rain forests. Ecological applications, 26(6), 1881-1895).

Thank you for raising this point. This paper and your other suggested reference are now included in the discussion (l. 515 in the track-changes version).

Reviewer 2 Report

I think this is a well written and interesting paper on the network properties of tree-fungal systems from a diversity experiment. The authors have utilised data from a well-designed study and have conducted a sensible and robust analysis of the network properties from the existing data.

I found the methods and results to be very clear when describing the complex analysis and I appreciated the use of figures to describe the analysis. Overall I think the findings are of interest, and I appreciated the authors use of a table in the discussion to help readers put these in context. I have a few minor suggestions that could improve the paper but overall I found it very good.

Specific comments:

Line 47-48 communities of what? Lines 59-61 references would be useful here Line 86 define C score and give the original reference A figure to describe the properties of nestedness, modularity, connectance and generality might be useful for readers unfamiliar with network analysis I am not clear on why all analyses were conducted with non-parametric tests, it would be useful to explain this choice Figure 2 – this figure would be more useful as boxplots to visualise spread around median Table 1 – describe the difference between “number OTUs” and “Fungal OTU richness” in the footnote Figure 3 – the horizontal bars in this plot and Figure 4 are not described. Perhaps consider a violin plot as an alternative, currently the plots are quite messy I don’t find Figure 5 particularly informative, it is not clear how fungal groups are arranged on the y axis nor what the taxonomic relationships of the trees on the x axis are. I’m not sure this figure is needed as it is mentioned only briefly Figure legends need more detail throughout to fully describe the figures Section 4.2 when you refer to hypotheses e.g. line 366 is it useful to remind the reader what these were The differences in phi coefficient between diversity groups are relatively small, although significant, and the variation within groups is also high. It is interesting that the variation is actually nearly as large in the monocultures as in the high diversity plots – might this reflect different tree species having different levels of associated specialisation? I don’t understand Figure S4 e.g. what are the x axes?

Author Response

I think this is a well written and interesting paper on the network properties of tree-fungal systems from a diversity experiment. The authors have utilised data from a well-designed study and have conducted a sensible and robust analysis of the network properties from the existing data. I found the methods and results to be very clear when describing the complex analysis and I appreciated the use of figures to describe the analysis. Overall I think the findings are of interest, and I appreciated the authors use of a table in the discussion to help readers put these in context. I have a few minor suggestions that could improve the paper but overall I found it very good.

Thank you for your interest in our analysis. We have edited the manuscript and in particular the figure legends according to your very helpful suggestions, as detailed below.

Specific comments:

Line 47-48 communities of what?

We have clarified this sentence to indicate tree diversity experiments.

Lines 59-61 references would be useful here

Thank you for this suggestion, we have added references for the most prominent applications of network analysis (l. 65 in the track-changes version).

Line 86 define C score and give the original reference

Thank you for pointing out this omission, we have edited the sentence accordingly.

A figure to describe the properties of nestedness, modularity, connectance and generality might be useful for readers unfamiliar with network analysis

We have added a supplementary figure that describe the network properties (Figure S1). We hope that this is helpful for some readers who are not familiar with the concepts, without cluttering the manuscript.

I am not clear on why all analyses were conducted with non-parametric tests, it would be useful to explain this choice

This is now mentioned in the methods section (l. 328 in the track-changes version). Distributions of the results of the subsampling approach are visible in the overplotted dots/histograms of the figures.

Figure 2 – this figure would be more useful as boxplots to visualise spread around median

Thank you for this suggestion. Due to the absence of our lead-author for health reasons, we cannot produce this figure at this moment. In accordance with the advice given by the editorial office, we hope our other revisions are to your satisfaction. We will supply this figure for the final version, if necessary.

Table 1 – describe the difference between “number OTUs” and “Fungal OTU richness” in the footnote

Thank you for this comment. We have edited the header of the table to clarify the distinction between the number of OTUs included in the network and the total number of detected fungal OTUs.

Figure 3 – the horizontal bars in this plot and Figure 4 are not described. Perhaps consider a violin plot as an alternative, currently the plots are quite messy

We have now explained the histograms in the legends, which are more detailed than a violin plot.

I don’t find Figure 5 particularly informative, it is not clear how fungal groups are arranged on the y axis nor what the taxonomic relationships of the trees on the x axis are. I’m not sure this figure is needed as it is mentioned only briefly. Figure legends need more detail throughout to fully describe the figures.

Thank you for pointing out the missing information, which are now explained in the figure legend. As the OTUs displayed in this figure are highly specialized, they are not shared by any of the tree species, independent of their taxonomic relationship. The tree species are therefore not arranged by taxonomy. A further comment on the figure has been introduced in the text.

Section 4.2 when you refer to hypotheses e.g. line 366 is it useful to remind the reader what these were

Thank you for the suggestion. We have repeated the hypotheses accordingly (l. 486 and 490 in the track-changes version).

The differences in phi coefficient between diversity groups are relatively small, although significant, and the variation within groups is also high. It is interesting that the variation is actually nearly as large in the monocultures as in the high diversity plots – might this reflect different tree species having different levels of associated specialisation?

You are correct that different numbers of specialized OTUs existed for the different tree species. We now mention this together with Figure 5 (l. 428 in the track-changes version). However, since the same tree species were examined at all diversity levels, the different numbers of specialized OTUs did not contribute to the differences in phi-coefficient between the diversity levels.

I don’t understand Figure S4 e.g. what are the x axes?

These graphs are histograms with the x-axes showing the number of trees where each OTU was detected in the subsample combinations. This is now mentioned in the in the figure legend (now figure S5) and in reference to the figure in section 3.2 .